

# The restoration of the endangered *Sambucus palmensis* after 30 years of conservation actions in the Garajonay National Park: genetic assessment and niche modeling

Priscila Rodríguez-Rodríguez[1], Alejandro G. Fernández de Castro[2] and Pedro A. Sosa[1]

[1] Instituto Universitario de Estudios Ambientales y Recursos Naturales (IUNAT), Universidad de Las Palmas de Gran Canaria, Las Palmas de Gran Canaria, Canary Islands, Spain
[2] Departamento de Biodiversidad y Conservación, Real Jardín Botánico de Madrid, CSIC, Madrid, Spain

## ABSTRACT

The translocation of individuals or the reinforcement of populations are measures in the genetic rescue of endangered species. Although it can be controversial to decide which and how many individuals must be reintroduced, populations can benefit from reinforcements. *Sambucus palmensis* is a critically endangered endemic to the Canary Islands. During the past 30 years, the Garajonay National Park (La Gomera) has carried out an intensive program of translocations using cuttings, due to the low germination rates of seeds. To assess the effect of the restorations on the population genetics of *S. palmensis* in La Gomera, we collected 402 samples from all the restored sites and all known natural individuals, which were genotyped with seven microsatellite markers. In addition, we conducted a species distribution modeling approach to assess how restorations fit the ecological niche of the species. Results show that there is a high proportion of clone specimens due to the propagation method, and the natural clonal reproduction of the species. Nonetheless, the observed heterozygosity has increased with the restorations and there still are private alleles and unique genotypes in the natural populations that have not been considered in the restorations. The population of Liria constitutes a very important genetic reservoir for the species. To optimize future reintroductions, we have proposed a list of specimens that are suitable for the extraction of seeds or cuttings in a greenhouse, as well as new suitable areas obtained by the species distribution models.

# INTRODUCTION

The preservation of endangered plant species usually involves population restorations, either by reinforcement of the extant populations or the reintroduction of new populations. Before starting restoration programs, biological information on the species must be first gathered in order to determine the most important factors limiting the growth of the

Corresponding author
Priscila Rodríguez-Rodríguez,
priscila.rodriguez@ulpgc.es

founding population (*Heywood & Iriondo, 2003*). The biological purpose of the restorations is to increase the species' survival prospects by recovering its evolutionary potential and autonomous ecological behavior (*Godefroid et al., 2011*). These measures often involve translocating genotypes across geographic ranges. This is a very controversial practice in which the need to maximize genetic diversity and avoid inbreeding depression is balanced against the maintenance of coadapted gene complexes (outbreeding depression; *Storfer, 1999*; *Hufford & Mazer, 2003*; *González-Pérez, Sosa & Batista, 2009*). In this respect, it has been extensively argued that increasing gene flow largely improves fitness and evolutionary potential of inbred populations, without high risks of outbreeding depression (*Frankham, 2015*). Nevertheless, many restoration programs have taken place without prior knowledge of the genetic background of the populations. The genetic variability within and between natural populations should be considered before starting reintroductions and translocations of genotypes. Also, many reintroductions are performed without analytical knowledge of the habitat and the autoecology of the species. Therefore, the selection of areas for reintroduction is often decided on the basis of intuitive or informal expertise, as the choice of unsuitable habitats is considered a frequent reason for failure (*Godefroid et al., 2011*).

Oceanic islands ecosystems are generally threatened either by biological (inherent to islands) or socio-economic factors. The biological causes are mainly habitat loss or degradation, small populations sizes and fragmentation, and the introduction of alien species or direct predation. The socioeconomic factors are mainly due to demographic and economic growth accompanied by a lack of natural resources management capacity, and with special relevance in the Canary Islands, the high touristic pressure (*Whittaker & Fernández-Palacios, 2007*; *Caujapé-Castells et al., 2010*). Climate change is also considered a major threat to the island flora, which could disrupt inter-specific mutualisms, or shift the elevation of cloud layers (*Harter et al., 2015*), very important for humid ecosystems, such as the laurel forest.

In Macaronesia, the laurel forest is one of the best representations of this insular vulnerability as it has experienced a major reduction since human colonization. In the Canaries, most of the laurel forest disappeared after the arrival of Castilians (15th century), affecting the distribution and abundance of many species (*De Nascimento et al., 2009*; *Fernández-Palacios et al., 2011*). Despite its slow recovery due to the abandonment of agricultural land, the extant laurel forest represents 18% of its original area (*Fernández-Palacios et al., 2011*). However, the forests in the Garajonay National Park in La Gomera, which have suffered less due to human colonization than in the other islands (*Nogué et al., 2013*), are considered to be the best relics of laurel forest. Indeed, the island holds all the laurel forest types present in the Canary Islands, with a high representation of endemic species (*Fernández-Palacios et al., 2017*). Therefore, La Gomera constitutes an optimal scenario to test habitat management initiatives.

The present study used *Sambucus palmensis* as a model species to evaluate the effectiveness of long-term conservation programs in clonal perennial species. Saúco or Canary elderberry is a rare endemic of the Canarian Archipelago and is present in four of the seven main islands (*Arechavaleta et al., 2010*): Tenerife, La Gomera, La Palma and Gran

Canaria. It falls within the category of "In danger of Extinction" in the Spanish Catalogue of Threatened Species and as "Endangered C2a" in the IUCN Red List (*Marrero et al., 2011*). In fact, it is one the tree species at highest risk of extinction in the Canary Islands. These considerations are due to the small number of naturally occurring individuals (<100) in the four islands. Other factors that could affect this threatened species are herbivory by goats and rats, habitat loss as well as forest fires and change in land use (*Fernández-López & Velázquez-Barrera, 2011*; *Marrero et al., 2011*). In fact, signs of a recent genetic bottleneck were estimated in two populations in La Palma and Tenerife (*Sosa et al., 2010*).

The latest census for *S. palmensis* in the Canary Islands confirmed that there exist 1,387 specimens, with a low percentage of natural origin. The origin of the individuals is confusing due to their connection to agricultural lands and cultivation, especially in Tenerife and La Gomera (*Marrero, Bañares & Carqué, 2015*) The highest number of individuals are currently found in La Gomera, (1090), distributed in more than 12 sites in the surroundings of the Garajonay National Park. Despite the high number of specimens, only 25 have been considered to be of natural origin, while the rest are the result of restoration programs developed over more than 30 years (*Marrero, Bañares & Carqué, 2015*). By the 1980s, the populations of La Gomera were reduced to a few individuals, which led to the urgent need of restoration activities (*Marrero, Bañares & Carqué, 1998*; *Marrero, Bañares & Carqué, 2015*). Later on, due to a fire that occurred in 2008 in La Gomera which cleared the laurel forest canopy, the locality of Liria went through a population expansion, from four known individuals to the 40 individuals that we analyzed in this article (*Fernández-López, Gómez-González & Gómez, 2014*). In the early stages of the conservation program, since germination rates were very low, the reintroduced specimens were obtained through vegetative propagation by cuttings (*Marrero, Bañares & Carqué, 1998*). Surprisingly, after these restorations, sexual propagation has been detected in climatically optimal years, increasing the size of the restored sites (*Marrero, Bañares & Carqué, 2015*). Consequently, individuals obtained by cuttings or descendants of old reintroductions have been translocated to other sites, possibly leading to a homogenization of the genetic structure. Although there has been a significant increase in the number of individuals, the genetic background has not been considered in any of the restorations. In the previous study of *S. palmensis* covering its entire distribution (*Sosa et al., 2010*), low genetic diversity, a high number of identical genotypes and overall exclusive alleles were found in La Gomera. A strong connection with Tenerife was also detected, possibly due to the anthropogenic transfer of individuals between islands. However, in the study by *Sosa et al. (2010)*, only three markers were polymorphic for La Gomera, which could have led to misinterpretation of the number of genotypes.

The aims of this study were (1) to estimate changes in genetic diversity after the restoration programs in La Gomera, (2) to identify the current genetic structure and the number of identical genotypes in the populations, (3) estimate the topoclimatic suitability of *S. palmensis* in La Gomera with natural and reintroduced occurrences and (4) to provide a background for future conservation programs in the Garajonay National Park.

## MATERIAL AND METHODS

### Study species

*Sambucus palmensis* Link, (*Sambucus nigra* subsp. *palmensis* (Link) R. Bolli) (Sambucaceae), It grows in the laurel forest vegetation zone between 600–1,000 m a.s.l., with a preference for shady and humid places. It can also be found on the margins of agricultural fields being grown by locals due to its historical use for medicinal purposes (*Beltrán et al., 1999*; *Abdala et al., 2014*). This tree can reach up to 4–6 m, is a hermaphroditic taxon and the fruits can be dispersed by birds (*Marrero et al., 2011*). Also, the species extensively propagates by vegetative reproduction, which has facilitated its reintroduction by cuttings. However, the low regeneration capacity of the species and the high mortality rates of new individuals leads one to think that *Sambucus palmensis* presents reproductive self-incompatibility and inbreeding (*Marrero, Bañares & Carqué, 1998*).

### Sample collection

The fieldwork was performed under the project 255/2011 funded by "Organismo Autónomo de Parques Nacionales" and the sample collection was assisted by the Garajonay National Park staff. Leaf samples from all the naturally occurring individuals (47), present in the sites of Acebiños, Ancón de Candelaria, Liria, Meriga and Presa de Las Rosas, and a significant representation of the reintroduced individuals from the whole distribution in La Gomera (355) were collected during 2012 and 2013. The categories of natural or reintroduced were considered according to the Garajonay National Park records and annotations. The individuals from Liria and Presa de Las Rosas are all natural, with no reintroduced individuals. In total, 402 samples were collected and genotyped for the study. They are distributed in 15 sites according to their geographical distribution or management area for the National Park. The distribution of the analyzed individuals and the sampling sites are represented in Fig. 1. Young leaves for all specimens were collected and stored in plastic bags with silica gel for their conservation.

### Microsatellite development and genotyping

In this study, seven SSR markers were used for the genotyping of *Sambucus palmensis* individuals in La Gomera. Five markers that were developed for *Sambucus nigra* (*Clarke & Tobutt, 2006*) and characterized for *S. palmensis* (*Sosa et al., 2010*) were tried for our set of samples. However, only three of these markers (EMSn017, EMSn025, EMSn003) yielded enough polymorphism in the La Gomera populations to be considered. Therefore, we developed four new microsatellite markers for *S. palmensis* to improve the accuracy of the genotypes, coded as Sam_Tet2, Sam_Hex2, Sam_Hex1 and Sam_Tri8 (Supplemental Information 1).

DNA was extracted from silica-gel-dried young leaves using a modified 2× CTAB protocol (*Doyle & Doyle, 1987*). DNA was subsequently purified with the commercial kit Gene Elute PCR Clean-up (Sigma Aldrich, St. Louis, MO, USA). After the characterization of the microsatellites, the whole set of samples was extracted with Invisorb DNA Plant HTS 96 KIT INVISORB.

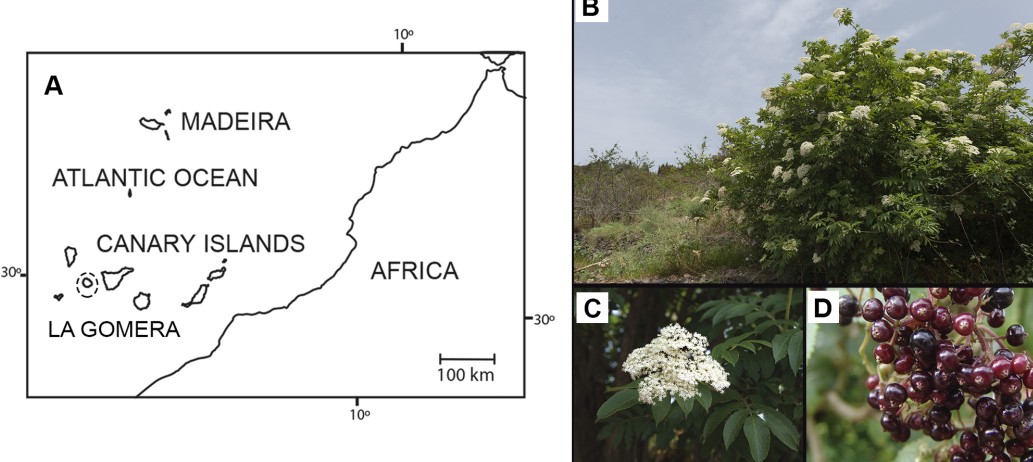

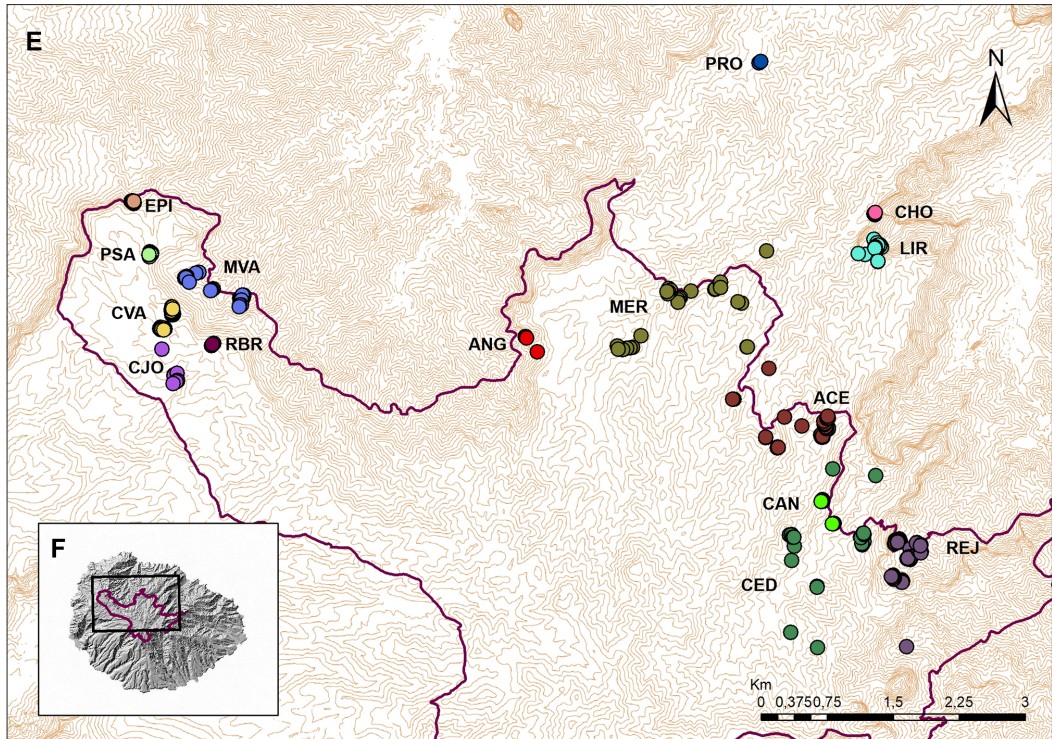

**Figure 1  Geographical distribution of *Sambucus palmensis* in La Gomera.** (A) Geographical situation of the Canarian archipelago. (B) Adult specimen of *Sambucus palmensis* (C) Details of the inflorescence. (D) Fruits. Image by J. Damián Esquivel Díaz, under a CC-BY-NC-SA license: http://www3.gobiernodecanarias.org/medusa/mediateca/ecoescuela/?attachment_id=2606. (E) Map of the distribution of the individuals sampled. The 15 areas described for population management are indicated, see the locality codes in Table 1. (F) La Gomera Island with the Garajonay National Park situation (purple line).

Microsatellite loci were selected from an Illumina paired-end shotgun library developed by the company AllGenetics (University of A Coruña) using three probe mixes of *S. palmensis*. We initially chose 20 primer pairs from this library of which 16 yielded some product and were labelled. Finally, four primer pairs amplified consistently with more than two alleles and were selected to complete the genotyping for all samples. For the initial testing, PCR was conducted individually with each primer pair in a 25 uL total reaction volume, which contained approximately: 20 ng of DNA 10 pmol of each primer, as well as PCR Master Mix until 25 uL were completed (Reddy-Mix, ABgene, Surrey, UK). Amplifications were performed using the following conditions: 3 min denaturation at 95 °C, 35 cycles of 30 s denaturation at 95 °C, 30 sannealing at 50–62 °C, and 72 °C for 1.5 min; followed by 5 min elongation at 72 °C. Reverse primers were color-labelled at the 5′-end with 6-FAM, PET, NED or VIC.

Once the new markers where characterized, we conducted the subsequent multiplex amplifications for the seven primer pairs using the QIAGEN Multiplex Kit (QIAGEN). PCR were performed in 15 µL reaction volumes: 7.5 µL of Multiplex PCR Master Mix, 1.5 µL primer mix (containing 0.2 µM of each primer in TE), 1.5 µl of Q-solution, 20–40 ng of DNA and $dH_2O$. Multiplexing was carried out in two primer groups as indicated in Supplemental Information 1. Following the manufacturer's instructions, PCRs consisted of a Touchdown protocol with the thermal conditions: 15 min at 95 °C, 10 cycles of 30 s at 94 °C, annealing for 90 s at 65 °C with a decrease of 0.5 °C per cycle and 60 s at 72 °C, followed by 20 cycles of 30 s at 94 °C, annealing for 90 s at 55 °C and 60 sat 72 °C, with a final extension of 30 min at 60 °C. All the products from both simple and multiplex PCR were detected on an ABI 3730 Genetic Analyzer and fragments were sized against the LIZ (500–250) size standard (Applied Biosystems, Foster City, CA, USA) and visualized using Genemapper 4.0 (Applied Biosystems, Foster City, CA, USA). We identified allele peak profiles at each locus and assigned a genotype to each individual.

## Statistical analysis

To estimate the incidence of clonality and identify the different genotypes for *S. palmensis* in La Gomera, multilocus matches were identified with GenAlex version 6.5 (*Peakall & Smouse, 2012*). Unique genotypes (only in one individual) were coded with consecutive numbers, and shared genotypes (in more than one individual) were coded with letters, from A to ZZ. With the same software, basic genetic diversity indices: average of alleles per locus (*Na*); number of private alleles (*Pa*), rare alleles (present in four sites or less; *Ra*), observed ($H_o$), and unbiased expected ($H_e$) heterozygosities for each locus and locality were estimated. Measures of allelic (*Ar*) and private allelic richness (*Par*) were calculated using HP-RARE 1.0 (*Kalinowski, 2005*), which uses rarefaction to correct for sampling error. To detect differences in genetic diversity before and after the restoration programs, the indices calculated for each locality were also estimated for the restored and natural individuals separately. Individual heterozygosity was also calculated to provide a list of the most suitable individuals for restoration programs, in order to increase the fitness of the founder individuals and their offspring (*Engelhardt, Lloyd & Neel, 2014*). On this list,

individuals with unique genotypes, a high individual heterozygosity and presence of rare alleles were identified.

Estimation of null alleles for each locality was carried out with MICROCHECKER 2.2.3 (*Van Oosterhout, Weetman & Hutchinson, 2006*). BOTTLENECK 1.2.02 software was used to identify any recent genetic drift events in the original set of individuals (*Cornuet & Luikart, 1996*). The two-phase mutation model (TPM), which is a modification of the stepwise mutation model (SMM), was implemented and shows to be a better fit for most microsatellite data sets (*Piry, Luikart & Cornuet, 1999*). In the TPM model, to optimize the most sensitive values for microsatellites, a proportion of SMM in the TPM = 0.000 and a variance of the geometric distribution for TPM = 0.36 were chosen.

Allele frequency information was analyzed using a nested analysis of molecular variance (AMOVA) (*Excoffier, Smouse & Quattro, 1992*) with ARLEQUIN software. The analyses were conducted with two different approaches, with the individuals being grouped either by locality or their origin (natural versus restored). In addition, a principal coordinate analysis (PCoA), using the covariance standardized method of pairwise codominant genotypic distances among individuals, was implemented with GENALEX version 6.5 (*Peakall & Smouse, 2012*).

To estimate the current genetic structure of the populations, all the genotypes were screened using a Bayesian admixture procedure with the software STRUCTURE (*Pritchard, Stephens & Donnelly, 2000*). The model was assumed to be of population admixture and correlated allele frequencies. 10 independent runs were conducted for each value of $K$ (from 1 to 15) and analysis consisted of a $10^5$ burn-in period replicated and a run length of $10^6$ replicates. The optimal number of clusters was found by the $\Delta$K method (*Evanno, Regnaut & Goudet, 2005*) visualized with STRUCTURE HARVESTER (*Earl & vonHoldt, 2012*). Results of 10 replicates of the best fit $K$ were processed using CLUMPP 1.1.2 (*Jakobsson & Rosenberg, 2007*) to determine the optimal clustering. The STRUCTURE HARVESTER results for the election of the optimal $K$ are presented in Supplemental Information 2.

## Species distribution modeling

The explanatory power of SDMs (Species Distribution Models) is often improved when accurate resolutions of spatial predictors are used (*Lassueur, Joost & Randin, 2006*; *Austin & Van Niel, 2011*). We therefore developed a set of accurate spatial climate layers at a 50-meter resolution, based on the network of meteorological stations of the whole archipelago (data provided by AEMET, http://www.aemet.es). Stations with 10 or more years of climate records were filtered for the models. For the monthly variables of minimum and maximum temperature and precipitation, a generalized additive model (GAM) with altitude, northness, latitude and longitude as predictor variables was developed. The best model was selected based on the AIC scores and then projected to La Gomera island. To account for spatial biases of the models, residuals of each model in each meteorological station were used to develop an interpolated map of residuals for each variable by ordinary kriging. This interpolated layer was added to the predicted value of the GAM model to obtain the final layers of each monthly variable. The final dataset of monthly variables was used to develop the bioclimatic variables described by *Hijmans et al. (2005)*, using

the 'dismo' package in R (*Hijmans et al., 2015*). Two topographic predictors: slope and topographic index (TPI), derived from the digital elevation model (DEM) of the archipelago were also incorporated.

A total of 441 reintroduced presence cells were recorded in total at 50 meter working resolution, 47 were natural presences in La Gomera and nine recorded in Tenerife and La Palma. A total of 51 presences were discarded as they could not be unambiguously assigned to natural or reintroduced populations in Tenerife and La Palma. To select climatic predictor variables for the modeling procedure first a correlation analysis was conducted for the values of the variables in the cells where the species was present with the R package 'ecospat' (*Broenniman et al., 2014*), which returned a value of four predictor variables. Then a PCA was conducted, and the 11 predictors which obtained the highest scores along the first three axes were retained. Finally, a hierarchical partitioning analysis was conducted (*Chevan & Sutherland, 1991*), with 'hier.part' R package (*Walsh & MacNally, 2013*), to select among those 11 variables, the five showing the highest independent contributions according to ecospat. These were bioclimatic variables 2, 10, 12 and 15 (temperature range, mean temperature of warmest quarter, annual precipitation, precipitation seasonality, respectively) and slope.

Three different niche modeling procedures were conducted with biomod2 package (*Thuiller et al., 2009*). The first one was calibrated with natural occurrences only, and the second one with both natural and introduced presences. Finally, the third one was calibrated with introduced occurrences only, to evaluate the model with the natural occurrences and obtaining therefore an evaluation of the introduction with independent data. In each of the three modeling procedures, five datasets were developed containing presence points and pseudoabsences were generated randomly only in unaltered habitats of the island. This was done to avoid generating pseudoabsences in potentially suitable areas were the species was not appearing due to habitat destruction instead of ecological constraints. In the third modeling procedure, pseudoabsences were also generated randomly but avoiding the areas where natural populations existed. In each of the three procedures the number of pseudoabsences differed: with natural occurrences, a total of 200 pseudoabsence points was generated, which were weighted in the models to account for the same importance as presences. With both natural and reintroduced presences, and with only introduced presences, the number of pseudoabsences matched the number of presences.

Six algorithms available in biomod2 were used: generalised linear models (GLM), generalised additive models (GAM), both with stepwise selection, boosting regression (GBM), multiadaptive regression splines (MARS), annual neural networks (ANN), and random forest (RF). 10 runs of each presence-pseudoabsence dataset were run for the first two modeling approaches. In each run, 85% of data was randomly selected for calibration and the rest for model evaluation. The third procedure was evaluated using the natural occurrences of the species. Models were evaluated by means of TSS and ROC scores. Any model with a score below 0.8 for any of the two metrics was excluded for the ensemble model. The remaining models were retained to build an ensemble model based on the contribution of each individual model weighted according to the TSS score.
**Table 1 *Sambucus palmensis* localities sampled in La Gomera.** The number of natural individuals, and number of genotypes per site are indicated. Unique genotypes are present in only one analyzed individual, while the shared genotypes were present in two or more individuals. Sites were grouped according to their geographical distribution or management area required by the National Park.

| Sampling site | Acronym | N | Natural individuals | Total number of genotypes | Shared genotypes | Unique genotypes | % of unique genotypes |
|---|---|---|---|---|---|---|---|
| Acebiños | ACE | 31 | 1 | 25 | 21 | 4 | 16.00 |
| Ancón de Candelaria | CAN | 17 | 1 | 6 | 5 | 1 | 16.67 |
| Angola | ANG | 5 | – | 5 | 4 | 1 | 20.00 |
| Cañada Jorge | CJO | 13 | – | 1 | 1 | 0 | 0.00 |
| Cordillera Vallehermoso | CVA | 25 | – | 22 | 16 | 6 | 27.27 |
| El Cedro | CED | 35 | – | 21 | 17 | 4 | 19.05 |
| El Chorrillo | CHO | 9 | – | 2 | 2 | 0 | 0.00 |
| El Rejo | REJ | 129 | – | 58 | 40 | 18 | 31.03 |
| Epina | EPI | 13 | – | 13 | 12 | 1 | 7.69 |
| Liria | LIR | 40 | 40 | 35 | 6 | 29 | 82.86 |
| Meriga | MER | 36 | 3 | 26 | 15 | 11 | 42.31 |
| Meseta Vallehermoso | MVA | 38 | – | 18 | 15 | 3 | 16.67 |
| Palo que salta | PSA | 7 | – | 7 | 5 | 2 | 28.57 |
| Presa Las Rosas | PRO | 2 | 2 | 2 | 0 | 2 | 100.00 |
| Raso de La Bruma | RBR | 3 | – | 3 | 1 | 2 | 66.67 |
| Average per site | | 26.9 | 9.4 | 16.2 | 10.6 | 5.6 | 31.71 |
| Total | | 402 | 47 | 147 | 63 | 84 | 57.14 |

**Notes.**

*N*, sample size.

## RESULTS

All microsatellite markers used for this study yielded enough polymorphism to identify the possible number of genotypes in the sampling sites analyzed and to detect possible identical genotypes. Out of the 402 individuals sampled, 147 different genotypes were found. 84 of these were unique genotypes (genotypes detected in only one individual), and 63 were shared genotypes (from more than one individual) (Table 1). We found some genotypes corresponding to a high number of individuals, for example the genotype JJ that matched 81 samples, the genotype V that matched 30 genotypes and the genotype UU matched with 15 individuals. The remaining genotypes matched between two and 10 individuals each. A detailed list of genotypes per locality is shown in Supplemental Information 3. The sampling site with the highest number of unique genotypes was the locality of Liria (29), where all individuals are natural, followed by the restored sites of El Rejo (18) and Meriga (11). The percentages of unique genotypes per locality were also higher in Presa de las Rosas (100%) and Liria (82.86%) than in the restored sites, which ranged from 0.00% to 66.67%.

Basic genetic diversity indices, such as allelic richness and expected heterozygosity values were similar across sites. The expected heterozygosity ranged from 0.357 (Presa de las Rosas) to 0.495 (Raso de la Bruma). The observed heterozygosity values did show higher differences across sites, ranging from 0.366 (El Cedro) to 0.714 (Cañada Jorge). Liria and

Table 2   Genetic diversity indices for *Sambucus palmensis* in La Gomera.

| Sampling site | N | Na | Pa | Ra | Ar | Par | $H_o$ | $H_e$ |
|---|---|---|---|---|---|---|---|---|
| Acebiños | 31 | 2.71 | – | 1.00 | 1.73 | 0.02 | 0.406 | 0.373 |
| Ancón de Candelaria | 17 | 2.86 | – | 2.00 | 1.95 | 0.04 | 0.605 | 0.491 |
| Angola | 5 | 2.14 | – | – | 1.87 | 0.00 | 0.600 | 0.467 |
| Cañada Jorge | 13 | 1.71 | – | – | 1.65 | 0.00 | 0.714 | 0.371 |
| Cordillera Vallehermoso | 25 | 2.57 | – | – | 1.88 | 0.02 | 0.606 | 0.466 |
| El Cedro | 35 | 2.57 | – | – | 1.71 | 0.01 | 0.366 | 0.368 |
| El Chorrillo | 9 | 2.14 | – | – | 1.74 | 0.00 | 0.698 | 0.415 |
| El Rejo | 129 | 2.86 | – | 2.00 | 1.93 | 0.05 | 0.549 | 0.479 |
| Epina | 13 | 2.14 | – | – | 1.89 | 0.00 | 0.692 | 0.486 |
| Liria | 40 | 2.29 | 3.00 | 5.00 | 1.73 | 0.26 | 0.421 | 0.376 |
| Meriga | 36 | 2.71 | 1.00 | 1.00 | 1.7 | 0.03 | 0.425 | 0.359 |
| Meseta Vallehermoso | 38 | 2.86 | – | 2.00 | 1.94 | 0.03 | 0.564 | 0.484 |
| Palo que salta | 7 | 2.29 | – | 1.00 | 1.71 | 0.03 | 0.469 | 0.364 |
| Presa Las Rosas | 2 | 1.57 | – | 2.00 | 1.57 | 0.23 | 0.500 | 0.357 |
| Raso de La Bruma | 3 | 2.14 | – | – | 1.93 | 0.00 | 0.667 | 0.495 |
| Average over pop | 26.86 | 2.37 | 0.27 | 1.07 | 1.80 | 0.05 | 0.552 | 0.423 |
| **All individuals** | 402 | 3.71 | – | – | – | – | 0.519 | 0.462 |
| **Reintroduced** | 355 | 3.14 | 2.00 | – | 2.82 | 0.16 | 0.532 | 0.462 |
| **Natural** | 47 | 3.43 | 4.00 | – | 3.43 | 0.77 | 0.426 | 0.402 |

Notes.

N, sample size; Na, average of alleles per locus; Pa, number of private alleles; Ra, rare alleles (present in 4 localities or less); Ar, rarefied allelic richness; Par, rarefied private allelic richness; $H_o$, observed heterozygosity; $H_e$, unbiased expected heterozygosity.

Meriga were the only sites with private alleles, and Liria presented the highest rarefied private allelic richness at 0.26 (Table 2). Moreover, eight sites presented rare alleles, which were present in four sites or less (Supplemental Information 4). Between restored and natural groups, the natural group displayed higher allelic richness and a greater presence of private alleles, but a lower observed heterozygosity than in the restored groups. The results of tests to detect recent bottleneck events in the natural individuals, considering them as a single population, were not significant for any of the tests implemented. Only the locus Sam_Hex2 presented evidence of null alleles in El Cedro, El Rejo and Meseta Vallehermoso.

AMOVA results were similar in the two approaches tested (Table 3). In both cases, the variance between individuals within groups was higher than that between groups. The variance between was 2.5% and 2.9%, between sites and between the natural and restored groups respectively. This low variance shows the lack of differentiation between the sitesstudied and between the natural and restored groups. As with the AMOVA results, the PCoA, with a total explanation of 62.06% did not reveal a clear aggrupation between natural and restored individuals (Fig. 2).

In the Structure analysis, we found two possible best values for $K$ according to the $\Delta K$ and the mean of log-likelihood values ($K = 2$ and $K = 5$) (Supplemental Information 2). Therefore, both possibilities are shown in Fig. 2. In $K = 2$, all individuals were admixed in
a)

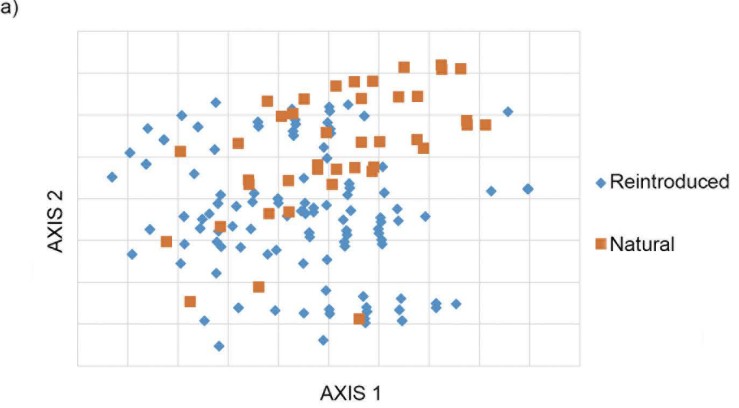

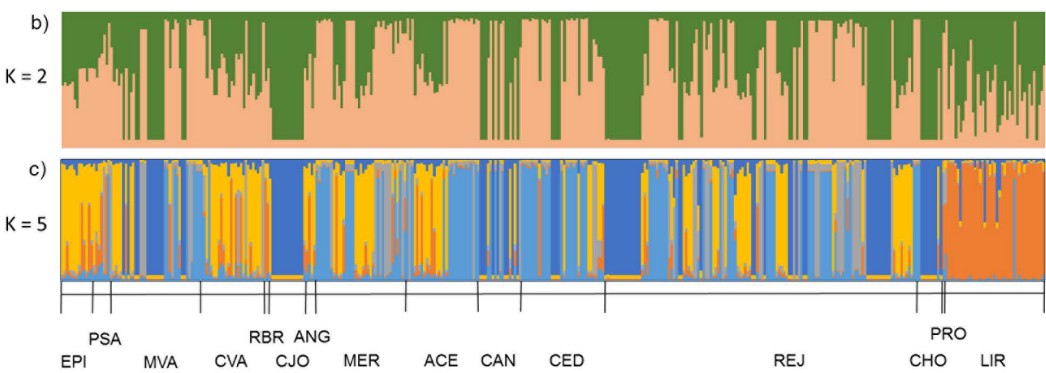

**Figure 2** **Genetic structure of *Sambucus palmensis* in the Garajonay National Park.** (A) Principal co-ordinate analysis (PCoA) for all *Sambucus palmensis* individuals sampled in the Garajonay National Park (La Gomera). The individuals were represented according to their origin (reintroduced or natural). The first two axes explained 62.06% of the total variation. (B and C) Bar plots for the proportion of coancestry inferred from Bayesian cluster analysis implemented on STRUCTURE and CLUMPP. (B) includes all the individuals grouped in $K = 2$ and (C) in $K = 5$ following the STRUCTURE HARVESTER results shown in Supplemental S2. Locality codes are indicated in Table 1.

**Table 3** **AMOVA analysis for *Sambucus palmensis* in La Gomera.** Individuals were grouped according to their sampling site and origin (natural or reintroduced).

| Source of variation | Degrees of freedom | Sum of squares | Variance of components | Percentage of variation | *F*-statistics |
|---|---|---|---|---|---|
| **All sites** | | | | | |
| Among groups | 14 | 14.7 | 0.012 | 2.5 | |
| Within groups | 789 | 370.0 | 0.469 | 97.5 | |
| Total | 803 | 384.6 | 0.481 | | 0.025[***] |
| **Natural vs. Reintroduced** | | | | | |
| Among groups | 1 | 2.9 | 0.014 | 2.9 | |
| Within groups | 802 | 381.8 | 0.476 | 97.1 | |
| Total | 803 | 384.6 | 0.490 | | 0.029[***] |

**Notes.**
[***]$P < 0.001$.

the two clusters described with evident lack of genetic structure. In the representation of $K = 5$, "Liria" was the only sampling site with a high assignation to a single cluster. The other sites presented admixture of the five clusters, except for "Cañada Jorge", but in that locality, all individuals shared the same genotype (Table 1).

Finally, we proposed a list with the 25 most suitable individuals with which to build a conservation genetic program to enhance the genetic variability of the populations. Individuals with unique or rare genotypes, which also presented a high individual heterozygosity and private or rare alleles were considered (Supplemental Information 5).

## Species distribution modeling

The PCA analysis conducted with all spatial predictors considered accounted for a 96.35% of the variance and showed patent differences between the niche of introduced and natural populations (Supplemental Information 6), corresponding to differences in the precipitation seasonality (bio04) and annual precipitation (Supplemental Information 7). Also, there were visible contributions of three intertwined variables: precipitation of the wettest month, precipitation of the wettest quarter and precipitation of the coldest month (variables 13,16,19); natural populations showed less temperature seasonality whereas they occurred in areas with lower precipitation (Supplemental Information 7).

The evaluations of the two modeling procedures differed significantly between occurrence datasets (GLMM, $F_{2,317} = 20.73$, $P \pm 3.575 \times 10^{-9}$, pseudoabsence dataset and model run as random effects) and models (GLMM, $F_{5,317} = 8.5499$, $P \pm 3.575 \times 10^{-9}$), with the introduced model showing better performance (Supplemental Information 8). In total, 66 models were not included with the natural dataset, 64 with the introduced one (evaluated against the natural occurrences) and 104 when considering all occurrences. Hence, the ensemble modeling was built with 85, 88, and only 51 respectively under the demanding quality threshold set for filtering models. However, the average TSS score for each modeling dataset was above 0.7, which is considered as fair (natural dataset, TSS = 0.79, introduced 0.78, all = 0.79) (*Guisan, Thuiller & Zimmermann, 2017*). Models in general showed therefore good performance, yet many of them were not included as they showed values around 0.7–0.8 score.

The suitable areas declared by the three modeling procedures projected suitable areas in the northern parts of the island in areas exposed to humid trade winds, but were conspicuously different (Fig. 3): the suitable areas for *S. palmensis* appeared in the northern-northeastern section of La Gomera when the model was calibrated with natural presences only. Models calibrated with introduced occurrences only showed on the contrary slightly more extended potential distribution for the species encompassing eastern regions. Regarding the comparison between natural calibration and the model calibrated with all ocurrences, the second showed lower suitability but also a lower threshold. Therefore, the whole model showed a higher extension. The potential distribution fell within protected areas only partially, with 1,345 cells inside protected areas out of 5,045 (26.83%). Only 17 cells did fall within the limits of Garajonay National Park.

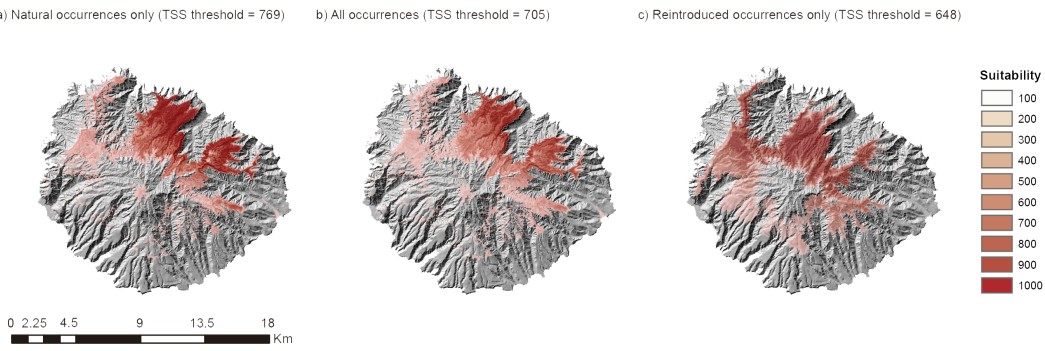

a) Natural occurrences only (TSS threshold = 769)   b) All occurrences (TSS threshold = 705)   c) Reintroduced occurrences only (TSS threshold = 648)

Suitability

0 2.25 4.5    9    13.5    18
Km

**Figure 3** Output maps of the ensemble model of topoclimatic suitability calibrated with: (A) natural occurrences only; (B) all occurrences; (C) introduced occurrences only.

## DISCUSSION

An integrative approach towards the conservation of endangered species, with the combination of molecular and modeling tools, is starting to be more considered, although few examples can still be found on islands (*Noël et al., 2010*; *Fernández-Mazuecos et al., 2014*; *Silva et al., 2015*). Our results suggest that the restoration programs of *Sambucus palmensis* in La Gomera have greatly improved the genetic status and distribution range of this species in the island. Although there is a high proportion of clonal specimens, natural regeneration has occurred in the restored sites, generating new genotypes and alleles that were not present in the original populations. Moreover, the restored sites are occupying new climatic suitable areas, which has led to an expansion of the distribution, compared to the natural populations. Nonetheless, there are still some major concerns in the conservation of *S. palmensis*, such as the difficulties in sexual reproduction, or the high mortality rates of young plants.

The genetic diversity estimates for La Gomera are higher than those reported by *Sosa et al. (2010)*. This is an expected result due to the increase in the number of polymorphic microsatellite markers. In the aforementioned study, only five markers were included, with 80% of polymorphism in La Gomera. Moreover, the high number of samples taken for this more detailed survey are a better representation of the real populations. Despite the new findings, the genetic diversity in La Gomera might possibly still be lower than in Tenerife and La Palma, whose populations have not yet been analyzed with the new set of markers.

Since the article by *Sosa et al. (2010)* was published, the knowledge of genetic diversity measured with microsatellite markers in oceanic endemics has significantly increased (*Takayama et al., 2015*; *White et al., 2016*; *Yang et al., 2017*). Therefore, we are now able to confirm that *S. palmensis* presents moderate levels of genetic diversity for a rare endemic (*Sosa et al., 2011*). Overall, outcrossing species present higher diversity than selfing species (*Hamrick & Godt, 1996*; *Pérez de Paz & Caujapé-Castells, 2013*). In comparison with other self-incompatible species, the rare endemics *Bethencourtia hermosae* (*Rodríguez-Rodríguez, Pérez de Paz Paz & Sosa, 2018*) or the highly threatened species of the genus *Commidendrum* from St Helena island (*Gray et al., 2017*) presented very low expected heterozygosity levels.

But those species are single-island endemics and habitat-restricted. *S. palmensis* is present in four islands and was probably even more widespread in the past (*Beltrán et al., 1999*; *Sosa et al., 2010*). In this respect, *S. palmensis* revealed similar heterozygosity values to other endemics such as the widespread dioecious palm *Phoenix canariensis* (*Saro et al., 2015*) or the laurel forest tree *Ilex canariensis* (*Sosa et al., 2013*).

## Effects of restoration activities on genetic diversity and structure

Because of the main propagation method, by the reintroduction of cuttings, we have detected a high presence of clonal specimens. Just like its relative *Sambucus nigra*, *S. palmensis* also presents natural vegetative reproduction (*Marrero, Bañares & Carqué, 1998*; *Bañares et al., 2004*). Therefore, natural clones can be found in Liria, which presented 17.14% of shared genotypes. But the percentage of shared genotypes in the other populations that have been restored is much higher, due to the extensive reintroduction of clone specimens. On the other hand, natural regeneration has been detected in some of the sites, with the appearance of new unique genotypes and private alleles in the restored sites. Nonetheless, the private alleles that are present in the restored sites could have come from natural individuals which are now dead and not included in this study. Therefore, the translocation of genotypes may be increasing the chances of sexual reproduction, as has already been detected in some of the restored sites (*Marrero, Bañares & Carqué, 2015*).

The percentages of variation found between populations are low for an outcrossing species (*Hamrick & Godt, 1996*), and they are also lower than that found by *Sosa et al. (2010)* among populations within islands (15%). The variation values detected are also lower than those found for the relative *Viburnum treleasi*, which has low variation among populations and also presents both sexual and clonal reproduction (*Moura, Silva & Caujapé-Castells, 2013*). Therefore, there is evidence that the admixture of genotypes across populations have favored the gene flow across the habitat range of *S. palmensis* in La Gomera. In the STRUCTURE results, Liria, which represents the best-conserved natural population in La Gomera, was the only locality that presented less admixture of individuals. The high number of unique genotypes and the presence of private alleles in Liria suggest that these individuals have been rarely used as a genetic source in the restoration programs, although some individuals were used for propagation before the expansion in 2008 (Á Fernández-López, personal comment, 2017). Although we did not find a sign of recent bottleneck events in the natural source, most of the individuals came from Liria, which has naturally increased its population size since the monitoring programs started. Thus, bottleneck events in the other natural sites are difficult to infer, as only a few individuals have remained.

Overall, there was a light increase in the observed heterozygosity in the restored sites. These results, together with the high admixture found in the genetic structure, also explain the artificial gene flow implemented with the restorations. Outcrossing of inbred isolated populations is playing a major role in the genetic rescue of endangered species (*Love Stowell, Pinzone & Martin, 2017*), but a balance between genetic rescue and "outbreeding depression" must be found in the management of populations, paying attention to the particular needs in each case (*Hufford & Mazer, 2003*). In addition, the habitat and climatic

continuity of the laurel forest in La Gomera, together with the outcrossing system of the species could have favored the gene flow between the past populations of *S. palmensis*, hindering high population differentiation or local adaptation. Even if the translocations of genotypes have led to outbreeding depression, the advantages of outcrossing can be greater, especially for self-incompatible species, as it increases the chances of finding available mates (*Willi et al., 2007*; *Pickup & Young, 2008*). Despite the increased observed heterozygosity, the number of alleles (the average of alleles per locus and allelic richness) are still lower in the restored individuals compared to the natural ones. These results can be taken as positive, because they indicate that the restored sites can still benefit from a greater outcrossing and admixture with the natural individuals within La Gomera.

## Insights from SDMs

This is the first modeling approach at such accurate scale developed for La Gomera. The development of three models based on different datasets for calibration allowed to draw two conclusions about the accuracy of the decision on the placement of the reintroductions. The PCA and the projections of the models clearly show differences in the ecological niche of the natural populations and the introduced ones, which survived successfully under other conditions. Natural populations were localized in areas with lower precipitation and allowed to predict only a part of the total potential area. From the discrepancy between the two niches, three main conclusions can then be drawn.

The first one is that the natural populations, which are very restricted, show a truncated niche, as they only reflected part of the ecological conditions of the species and many individual models did not reach the quality threshold to be included. From the conservation perspective, this is a frequent episode for threatened species witnessing the reduction of the realized niche, and a challenge for modeling procedures which normally should assume niche equilibrium This is the first case in the Canaries of assessment of niche filling of a species through practical reintroduction. Indeed, this kind of assessments supported by independent presence records are rather infrequent (*Guisan, Thuiller & Zimmermann, 2017*). Therefore, the second conclusion is that the selection criteria of enclaves for reintroduction was appropriate, given the physiological performance of individuals, and despite the climatic conditions did not match those of the scarce natural remaining populations. The quality of the models that considered introduced occurrences, assessed either with independent data or not, was significantly higher than considering natural occurrences only, with few models failing. The fairly higher number of introduced occurrences on the other hand can explain why the potential distribution predicted by the model calibrated with introduced occurrences and absences within La Gomera is higher than any other model. Finally, models served as a tool to identify further areas for protection. At present, protected areas other than Garajonay National only cover a quarter of the potential distribution.

## Recommendations for conservation actions

One of the main purposes of restoration ecology is to simulate the characteristics of the natural populations (*Pavlik, 1996*) and the restored sites do not show diminished

levels of genetic diversity compared to the original populations, despite the high number of clonal specimens. Also, SDMs support that reintroduced specimens match properly missing ecological conditions in the remaining populations. However, to improve the sexual regeneration in future reintroductions, further studies of the reproductive biology of *S. palmensis* are encouraged. The detection of the possible causes of self-incompatibility would help to increase the level of available mates and therefore gene flow and offspring. Also, more demographic studies such as that carried out in Meseta Vallehermoso (*Marrero, Bañares & Carqué, 2015*), will help to monitor the fitness and survival of restored sites over time. The combination of demographic and genetic studies is vital to ensure the recovery of endangered species (*Oostermeijer, Luijten & Den Nijs, 2003*).

As an urgent measure to maintain the genetic diversity of *S. palmensis* in La Gomera, we have already provided a list to the Garajonay National Park managers with the best candidates for a conservation genetic program (Supplemental Information 5). We also believe that it is important to avoid the genotypes detected in this study that have been extensively used in some sites. As indicated in *Vergeer, Sonderen & Ouborg (2004)*, we would also suggest increasing the number of unrelated seed producer individuals to create sustainable and viable populations, which would also avoid inbreeding processes. Since propagation by cuttings is the most viable way of reintroducing new individuals, the consideration of all individuals with unique genotypes for future reintroductions is also a conservation measure to be taken into account. Although it is possible that individuals from Tenerife were introduced in the past (*Sosa et al., 2010*), we consider that there is enough genetic variability in La Gomera to continue the restoration programs using the genotypes that are currently present on the island. Moreover, the suitability model based on the whole dataset of occurrences is a valid tool to identify other suitable areas for further reintroductions or translocations. The fact that the current *S. palmensis*' populations in La Gomera are within a National Park figure, clearly has benefit the success of the restoration programs, in respect to the populations in other islands. Although we have found suitable areas out of Garajonay that could be useful in future reintroduction attempts. Therefore, it is important to take into consideration endemic rarity in the design of protected areas (*Irl et al., 2017*).

It has been detected that good germination rates and seedling establishment highly depends on years of good precipitation (*Marrero, Bañares & Carqué, 2015*). But the long survival of individuals is determined by the clearance of the forest, reason why the Liria population expanded after the fire in 2008. Although introduced herbivores are a major threat for the island endemic flora (*Reaser et al., 2007*; *Garzón-Machado et al., 2010*; *Irl et al., 2014*), the laurel forest is not inhabited by rabbits, and the Garajonay National Park tries to control the effects of feral goats and sheep. However, predation of the fruits and leaves by rats has been observed (Á Fernández, personal comment, 2017). In this sense, we consider of importance to favor the establishment of new seedlings by manually clearing the canopy forest in the surroundings of *S. palmensis* individuals and a thorough control of invasive herbivores.

On a long-term basis, this case study will provide a great deal of information regarding the consequences of restoration actions in self-incompatible clonal species. Moreover,

these results can serve as a guideline for the restoration programs in Tenerife, La Palma and Gran Canaria, whose island governments are also restoring *S. palmensis* populations. For example, in Gran Canaria, only two naturally occurring individuals and some cultivars were found prior to the reintroductions. Therefore, a better knowledge of the genetic background of the restored individuals and the climatic suitability of the species would increase the success and long survival of the populations.

## ACKNOWLEDGEMENTS

We thank Ángel Fernández, Sito Chinea and Ángel García from Garajonay National Park and the colleagues Agustín Naranjo Cigala, Claudio Moreno Medina, Juan José Robledo Arnuncio, Miguel Ángel González, Pedro Luis Pérez de Paz and Leticia Curbelo for the help in the samples collection and the laboratory work.

### Funding

This research was funded by ''Organismo Autónomo de Parques Nacionales'' (project 255/2011). Priscila Rodríguez Rodríguez received a fellowship from ''Agencia Canaria de Investigación, Innovación y Sociedad de la Información'' to finance her PhD. The funders had no role in study design, data collection and analysis, decision to publish, or preparation of the manuscript.

### Grant Disclosures

The following grant information was disclosed by the authors:
Organismo Autónomo de Parques Nacionales: project 255/2011.
Agencia Canaria de Investigación, Innovación y Sociedad de la Información.

### Competing Interests

The authors declare there are no competing interests.

### Author Contributions

- Priscila Rodríguez-Rodríguez conceived and designed the experiments, performed the experiments, analyzed the data, prepared figures and/or tables, authored or reviewed drafts of the paper, approved the final draft, population genetic analysis.
- Alejandro G. Fernández de Castro performed the experiments, analyzed the data, prepared figures and/or tables, authored or reviewed drafts of the paper, approved the final draft, species distribution modeling.
- Pedro A. Sosa conceived and designed the experiments, contributed reagents/materials/analysis tools, authored or reviewed drafts of the paper, approved the final draft, principal Investigator.

### Field Study Permissions

The following information was supplied relating to field study approvals (i.e., approving body and any reference numbers):

Field experiments were approved by the Garajonay National Park under the supervision of the Head Manager Ángel Fernández López.

## DNA Deposition

The following information was supplied regarding the deposition of DNA sequences:

The newly developed microsatellite markers described here (Sam_Tet2, Sam_Hex2, Sam_Hex1, Sam_Tri8) are accessible via GenBank accession numbers LT600693, LT600694, LT600695 and LT600696.

## Data Availability

ResearchGate: https://www.researchgate.net/publication/325204178_The_restoration_of_the_endangered_Sambucus_palmensis_after_30_years_of_conservation_actions_in_the_Garajonay_National_Park_genetic_assessment_and_niche_modeling_Supplemental_S9_Microsatellite_raw_data.

## Supplemental Information

Supplemental information for this article can be found online at http://dx.doi.org/10.7717/peerj.4985#supplemental-information.

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
