# Peer review of "The restoration of the endangered Sambucus palmensis after 30 years of conservation actions in the Garajonay National Park: genetic assessment and niche modeling"

_PeerJ, doi:10.7717/peerj.4985_

## Round 0.1 · original submission · Minor Revisions

Reviewers and I have now read your manuscript and generally liked it as a very nice example of population genetics analysis applied to the conservation of an endangered species.

However, there are several issues which should be addressed for improving the work. In particular, Rev#2 noted that a more comprehensive sampling of the species throughout the archipelago would be much more interesting for the broader island biology community. I agree with this idea, but, at the same time, I understand that this could be very time- and work-demanding. So, it’s up to you if follow or not this recommendation. However, I concur with Rev#1 when they say that the discussion is very Canary-centric: a broader the discussion by also using literature from other archipelagos that encounter similar issues may draw interest of a wider readership.

Reviewer 1 ·

Basic reporting

Abstract
L21: Please delete “resolution”

Intro
L53-55. Whittaker & Fernandez-Palacios (2007) is a very general reference. Maybe think about using more specific references such as Caujapé-Castells et al. (2010) here instead. Also, a future threat for range-restricted endemic species on islands might be climate change (see e.g. Harter et al. 2015). This could be mentioned here.

Caujapé-Castells, J., Tye, A., Crawford, D. J., Santos-Guerra, A., Sakai, A., Beaver, K., Lobin, W., Florens, V. F. B., Moura, M., Jardim, R., Gómes, I. & Kueffer, C. (2010) Conservation of oceanic island floras: present and future global challenges. Perspectives in Plant Ecology, Evolution and Systematics, 12, 107-129.

Harter, D. E. V., Irl, S. D. H., Seo, B., Steinbauer, M. J., Gillespie, R., Triantis, K. A., Fernández-Palacios, J. M. & Beierkuhnlein, C. (2015) Impacts of global climate change on the floras of oceanic islands - projections, implications and current knowledge. Perspectives in Plant Ecology, Evolution and Systematics, 17, 160-183.


L66-76. I think this section (describing the study species) belongs in the methods with a separate heading (“Study species”).

L66-68. Please give a reference for this statement.

L83 “recent bottleneck”. What kind of bottleneck do you mean? A genetic bottleneck? Please be more specific here because this could mean multiple things.

L90-93. Out of interest: It seems like disturbances (such as fire) increase the reproduction success of S. palmensis. Would this be a possible conservation approach for future conservation efforts to increase the genetic diversity of the populations?


Figures
As you’re only dealing with a single species, I think one (or several) pictures if S.p. would help to visualize what kind of species it is and what sort of habitat it occupies. This could be included into Fig. 1.

Fig1.
Inlay A. Most people won’t know where or what La Gomera is. I would suggest to include a map of the North Atlantic and Canary Islands to precisely locate where the island lies.
Also, is it necessary to show other protected areas besides the national park? As far as I recall the other PAs are not mentioned in the MS and all these colors make things confusing.


Fig2.
Maybe considering adding a simple plot of the temperature and precipitation values of all natural and reintroduced individuals. You could draw convex hulls around their distributions to visualize the differences between natural and reintroduced. Or is this meant by 2a? However, from the caption and the figure it is not clear what the PCoA is showing and what the axes mean.

Fig3.
“Countor lines” I can’t see any counter lines in the maps. They are obviously too thin or the maps too small.
“whether the species is present or not”. SDMs do not show presence or absence of species. They show the environmental suitability.
3b). I was wondering why the SDM of all occurrences shows a smaller and more confined distribution than the subset only using reintroduced occurrences. This seems weird to me and is totally counterintuitive. Wouldn’t a larger set of occurrences with a wider environmental distribution lead to a larger area of suitability? Can you verify that this map is correct? If yes, can you discuss this issue in the discussion?

In general, all labels are too small in Fig3. Please increase label size! Also, choosing red and green as colors for the points of reintroduced and natural occurrences does not make sense, if your suitability map also includes red and green (in addition, 10% of men are colorblind with regard to red/green!). I cannot recognize the green points on the map. Maybe it would be better to have a regular digital elevation model (or something similar) to show the points and leave out the points in the SDM maps. These four maps could be combined to a 2x2 panel making all of the maps bigger in the final publication.

Experimental design

Material and methods
L114. “localities”. I think what you mean here is “sites”. I assume that you are using the direct translation from Spanish “localidades” but this does not sound right in English. Please check this throughout the MS.

L208. “projected TO La Gomera island”

L216. “402 presence cells”. As far as I understand you only use occurrence data from La Gomera, although the species has a wider distribution (Tenerife, La Palma, Gran Canaria) to model the distribution of S.p., right? This might lead to a truncation of the species’ distribution (by ignoring the distribution of S.p. on other islands which could differ in environmental conditions than the occurrences on La Gomera). This might result in a biased (or skewed) SDM because not the full niche of the species is considered. Is data from other islands available to model the complete distribution of S.p.? I assume the Atlantis database has a too coarse resolution for your approach. If additional data is not available, could you at least discuss this issue in the discussion. You already mention possible truncations there and I think this aspect should be mentioned as well.


Results
L283. Better write “we found two possible best values for K according…”

L316-318. Sentence unclear. Please rephrase!

Validity of the findings

Discussion
In general the discussion is very Canary-centric! To draw interest of a wider readership, I suggest to broaden the discussion by also using literature from other regions, at least from other archipelagos that encounter similar issues.

L325. “or high mortality rates of young plants” Does this have to do with introduced herbivores? Introduced herbivores selectively browse juvenile endemics because they are most palatable to them. If yes, you could expand on this issue, e.g. by citing the large body of literature that exists for the Canary Islands:

Cubas, J., Martín-Esquivel, J. L., Nogales, M., Irl, S. D. H., Hernández-Hernández, R., López-Darias, M., ... & González-Mancebo, J. M. (2017). Contrasting effects of invasive rabbits on endemic plants driving vegetation change in a subtropical alpine insular environment. Biological Invasions, 1-15.

Irl, S. D. H., Steinbauer, M. J., Messinger, J., Blume-Werry, G., Palomares-Martínez, Á., Beierkuhnlein, C., & Jentsch, A. (2014). Burned and devoured-introduced herbivores, fire, and the endemic flora of the high-elevation ecosystem on La Palma, Canary Islands. Arctic, antarctic, and alpine research, 46(4), 859-869.

Garzón-Machado, V., González-Mancebo, J. M., Palomares-Martínez, A., Acevedo-Rodríguez, A., Fernández-Palacios, J. M., Del-Arco-Aguilar, M., & Pérez-de-Paz, P. L. (2010). Strong negative effect of alien herbivores on endemic legumes of the Canary pine forest. Biological Conservation, 143(11), 2685-2694.

Irl, S. D. H., Steinbauer, M. J., Babel, W., Beierkuhnlein, C., Blume‐Werry, G., Messinger, J., ... & Jentsch, A. (2012). An 11‐yr exclosure experiment in a high‐elevation island ecosystem: introduced herbivore impact on shrub species richness, seedling recruitment and population dynamics. Journal of Vegetation Science, 23(6), 1114-1125.


L410ff. “Recommendations for conservation actions” Although I like this section and it offers some good suggestions and direct support for conservation, I somewhat miss a discussion on how protected areas are able to conserve and restore rare endemics. For example, a recent study from La Palma showed that protected areas generally harbor more rare endemics than non-protected areas (Irl et al. 2017). You could test, for example, how much of the suitable area calculated by the SDM lies within and without of protected areas to see if conservation goals are realistic or not.

Irl, S. D.H., Schweiger, A. H., Medina, F. M., Fernández‐Palacios, J. M., Harter, D. E.V., Jentsch, A., ... & Beierkuhnlein, C. (2017). An island view of endemic rarity—Environmental drivers and consequences for nature conservation. Diversity and Distributions, 23(10), 1132-1142.

Additional comments

Rodriguez-Rodriguez et al. et al. present an interesting study on the restoration of a highly endangered tree species from the Canary laurel forest (Sambucus palmensis) using a population genetics as well as a species distribution approach on the island of La Gomera. They analyze natural and restored populations, model the environmental niche of the species to assist conservation and extract some very concrete information for future conservation/restoration efforts (by providing a list of candidate individuals for future cuttings). All in all, I like the combination of modern techniques to assist ongoing conservation/restoration efforts. However, I have some issues that should be clarified before proceeding to publication. I hope you find my comments helpful, however I have to state that I’m not an expert on population genetics so I cannot comment this part (especially the methods).

·

Basic reporting

Rodriguez-Rodriguez and colleagues performed a population genetics analysis to infer the genetic diversity of one of the most endangered Canary island endemic plants, Sambucus palmensis, on the island of La Gomera. This plant species has been target species of conservation management for the past 30 years, however, the management mostly consisted of vegetative propagation of a few individuals, leading to concerns that there could be a serious risk of genetic impoverishment and inbreeding, eventually leading to the extinction of the species. The sampling included all 402 individual plants known in the wild on La Gomera (47 natural and 355 planted). Analysis of seven microsat regions revealed indeed a high proportion of clones but there is still a relatively high genetic diversity in the total population, which can be used - in combination with niche modelling results - for planning of future management strategies.
The study is well structured and written in good English. The introduction is adequate and gives sufficient background for the general readership of PeerJ. The only information I could not find is an estimate of the total number of individuals of Sambucus palmensis in the wild (all four islands). The figures are fine but again, I would prefer a distribution map including all four islands, instead of only La Gomera. All the raw data seems to be available, but I could not find a research permit number, even though most (all?) of the study was performed in a protected area, where such permits are certainly required. Please add this important information.

Experimental design

The authors report original primary research addressing an interesting question: can we maintain small populations of endemic plants even if we have to rely mostly on vegetative propagation. This is a pressing question on islands like the Canaries, where a lot of time and money is invested in such programs. The methods are described in sufficient detail and seem adequate. The only surprise to me is that no material from the other three islands with Sambucus palmensis populations (La Palma, Tenerife, Gran Canaria) was included, even though Tenerife and La Palma populations were already focus of a previous study (Sosa et al. 2010). Since it seems to be very rare on those other islands, this should be an easy task (in terms of additional samples) and would make the study much more comprehensive and meaningful. With samples from all islands, we would know the genetic diversity of the species as a whole (not only the La Gomera population), it would be possible to compare values between islands and most importantly, the species distribution modelling would make much more sense.

Validity of the findings

The findings are robust and discussed in a meaningful way. The suggestions for the species management, especially the list of individuals to be used for propagation, will be very useful for the Garajonay National Park staff. However, they will not be very useful for management of the S. palmensis populations on the other islands and we cannot infer, how useful (or not) it could be to transfer plants between islands.

Additional comments

In general, this is a nice and useful study. However, in my opinion, a more comprehensive sampling with Sambucus palmenis individuals from all 4 islands would be much more interesting for the broader island biology community. Focussing on La Gomera alone seems a bit strange to me and doesn't give us the big picture needed for a really holistic approach to long term management of the species.
I don't know how much additional effort would be needed to include those other populations (and how many individuals there are on the other islands, Gran Canaria only 2, Tenerife: ?, La Palma: ?) but the study would definitely benefit a lot.

---

## Round 0.2 · accepted · Accept

I think you have adequately replied to reviewers' remarks. I appreciate and agree that an extension of your work to other islands is not realistically feasible, although potentially interesting, in a short term. The explanation of the choice of La Gomera as a unique national Park hosting S. palmensis makes sense to me.

#